# Bioinspired Interfacial Friction Control: From Chemistry to Structures to Mechanics

**DOI:** 10.3390/biomimetics9040200

**Published:** 2024-03-27

**Authors:** Yunsong Kong, Shuanhong Ma, Feng Zhou

**Affiliations:** 1State Key Laboratory of Solid Lubrication, Lanzhou Institute of Chemical Physics, Chinese Academy of Sciences, Lanzhou 730000, China; yunsongkong@licp.cas.cn (Y.K.); zhouf@licp.cas.cn (F.Z.); 2College of Materials Science and Opto-Electronic Technology, University of Chinese Academy of Sciences, Beijing 100049, China

**Keywords:** friction control, lubrication regulation, chemistry, surface structure, mechanics

## Abstract

Organisms in nature have evolved a variety of surfaces with different tribological properties to adapt to the environment. By studying, understanding, and summarizing the friction and lubrication regulation phenomena of typical surfaces in nature, researchers have proposed various biomimetic friction regulation theories and methods to guide the development of new lubrication materials and lubrication systems. The design strategies for biomimetic friction/lubrication materials and systems mainly include the chemistry, surface structure, and mechanics. With the deepening understanding of the mechanism of biomimetic lubrication and the increasing application requirements, the design strategy of multi-strategy coupling has gradually become the center of attention for researchers. This paper focuses on the interfacial chemistry, surface structure, and surface mechanics of a single regulatory strategy and multi-strategy coupling approach. Based on the common biological friction regulation mechanism in nature, this paper reviews the research progress on biomimetic friction/lubrication materials in recent years, discusses and analyzes the single and coupled design strategies as well as their advantages and disadvantages, and describes the design concepts, working mechanisms, application prospects, and current problems of such materials. Finally, the development direction of biomimetic friction lubrication materials is prospected.

## 1. Introduction

Friction, the process of energy dissipation when two surfaces slide relative to each other, can be found everywhere in daily life and industrial manufacturing. On the one hand, friction plays a vital role in everyday life and production; on the other hand, friction causes severe wear and tear phenomena and requires colossal energy consumption. Therefore, it is necessary to develop various lubricant materials to regulate interfacial friction in specific situations. People have studied friction extensively and intensively for sustaining industrial production, conserving energy, and improving the quality of life, which has continued to drive the development of mechanical and materials science. However, with the improvement of people’s quality of life and rapid technological innovation, traditional lubrication materials sometimes find it challenging to meet the requirements of specific friction systems, which requires us to propose new strategies for controlling interfacial friction.

The diversity of life in nature showcases the beauty and functionality of matching form and purpose across all scales. The unique structures that have evolved in organisms due to common materials or specific physiological processes can inspire us to design materials, devices, or processes with desirable functions, which is the fundamental concept behind “bionics.” Over 3.8 billion years, a wide range of natural organisms have evolved organs and structures that can be adapted to complex operating conditions, including a wide range of ingenious friction and lubrication systems. Many of these organisms realize a wide range of tribological properties through different interfacial chemistry, surface structures at various scales, and mechanical properties of the biological structures to achieve the desired lubrication effect in a long-lasting and efficient manner. By further understanding how the complex functionalization and modulation of biological structures can be achieved, we can optimize the performance and realize the intellectualization of materials.

Humans have long noticed the excellent tribological properties of various organisms’ internal and external physiological structures and have conducted a series of related studies. From the perspective of solid surface lubrication, organisms in nature exhibit three main types of friction regulation strategies (Figure 1). One is the particular chemical nature of the surface, which realizes lubrication through the macromolecular layer on the surface of organisms with unique functions or the secretion of chemical substances with lubricating effects, such as the mucus secreted by the plant [1] and the synovial fluid and cartilage layer of mammals [2]. The second is the formation of structures on surfaces at various scales, such as the arrays of gecko feet [3] and the grooves on the surface of shark skin [4]. The third is to change the mechanical properties of the surface or subsurface to drastically alter the friction state at the interface, such as the hardening of the dermis of the sea cucumber and the contraction of fish muscles [5,6]. In the face of complex environmental conditions, it is often challenging to design biomimetic lubrication materials based on a single strategy to cope with the wide range of influencing factors in real situations, so researchers usually need to couple multiple strategies for material development.

This paper introduces the common forms of bio-lubrication modulation in nature and the corresponding application of biomimetic materials in friction systems from the standard lubrication systems in nature. This paper introduces the mechanisms of biomodulation of interfacial friction from three perspectives, namely, interfacial chemistry, surface structure, and surface mechanics, respectively, and analyzes the advantages and disadvantages of various biomimetic strategies, discusses the possibilities and superiority of multi-strategy coupling, and looks forward to the direction of the development of biomimetic interfacial friction modulation and the prospects for its application.

## 2. Surface Chemistry-Dominated Friction

It has long been noted that many plants and animals in nature can achieve lubrication effects through good hydration of their secretions or soft tissue surfaces. Jacob Klein, a famous tribologist, proposed the concept of hydration lubrication, described the role of the hydration layer in water lubrication, and explained the principles of many biological lubrication systems [7]. The water molecule appears to be electrically neutral. However, due to the dipoles caused by residual charges on the hydrogen and oxygen atoms, the water molecules will form a hydration layer around the polar groups (Figure 2). The hydration charges will repel each other when they are close, making it difficult for the hydration layer to overlap [8,9]. During aqueous lubrication, the charged groups at the interface can immobilize the oppositely charged hydrated groups during sliding via strong electrostatic interactions, meaning that the hydrated layer also reduces interfacial friction under high normal pressures, which is consistent with the working conditions in many cases in living organisms [10,11].

Researchers have studied biological structures with good lubricating properties and found that the mucus or surface with lubricating functions in plants and animals usually has a special chemical composition. For example, the components of plant secretions that play a lubricating role mainly include well-hydrated macromolecules such as polysaccharides and cellulose. One of the strategies for developing new lubricants is to analyze the mucus by extracting specific components or designing based on its composition. The mucilage in aloe vera is a suitable polysaccharide water-based bio-lubricant. Aloe leaves are rich in mucilage, whose main component is polysaccharides. Xu et al. [12] investigated the tribological properties of aloe mucilage and found that the mucilage can exhibit friction consistent with thin-film lubrication. Hakala et al. [13] extracted mucilage with a lubricating effect from fresh papaya fruit (Figure 3a–c), and the combination of nanofibers and water-soluble polysaccharides can form a gel-like structure. Arad et al. [14] evaluated the sulfated polysaccharide obtained from the red microalga Porphyridium sp., which showed good lubrication properties in rheological studies. Li et al. [15] reported the excellent lubricating properties of Brasenia schreberi mucilage (Figure 3d–f), in which there are a large number of polysaccharide cross-linked nanosheets, which can be combined into a solid polysaccharide layer on the glass surface through hydrogen bonding and the adsorption of a large number of water molecules during the lubrication process, and they form a hydration layer between the layers in order to effectively reduce the friction. The plant secretions mentioned in this paragraph and their tribological properties have been summarised in Table 1.

Compared to the limited lubricating properties of plant mucus, the lubrication system in animals usually maintains a lower COF and efficient lubrication under more complex and demanding conditions, as required for the proper functioning of various functions. In the human body, biological lubrication plays a role in almost every organ and tissue in the body all the time, such as the blinking lubrication by the tear fluid between the cornea and the eyelids [16], the lubrication of the esophagus by mucus containing biomolecules when swallowing food [17], the boundary lubricant film formed by salivary proteins in the oral cavity [18], and the synergistic lubrication of synovial fluid and cartilage in the joints [19]. Among them, the human joint lubrication system has been widely studied because of its close correlation with people’s quality of life and its excellent lubrication performance, which can work normally with a very low COF under high load conditions and shows excellent lubricating and anti-wear properties [20]. The synergistic effect of synovial fluid and structurally specialized cartilage in the joint system contributes to the excellent and stable lubricating properties.

The main components of synovial fluid include hyaluronic acid (HA), polyproteoglycans, and lubricin (Figure 4) [21]. HA is a high-molecular-weight linear polysaccharide that can bind many water molecules and separate the cartilage on both sides of the joint during sliding, which is essential for increasing synovial fluid viscosity [22]. At the same time, HA binds to phospholipids to anchor to the vesicle surface, and the combination of the two dramatically improves the hydration properties of synovial fluid. Polyproteoglycans have a natural hierarchical bottle-brush structure, with a backbone capable of forming interconnections or adsorbing onto the cartilage surface and hydrophilic glycan side chains capable of binding to water molecules [23]. Lubricin is also a glycoprotein with a bottle-brush structure that can act as a protective agent for chondrocytes. Klein et al. [24,25] explained the mechanism in detail for the specific form of action in polymer brush joint lubrication. Hydrophilic macromolecules contract in the dry state, ionize to form high osmotic pressures when hydrated, and maintain a stretched and swollen morphology, which prevents interfacial contact and resists applied loads [26,27]. At the same time, the hydration of polymer brushes causes them to aggregate at the sliding interface to form a boundary lubrication layer, further reducing friction [28].

Researchers have discovered or synthesized many macromolecular bio-lubricants with excellent properties based on understanding the lubrication mechanism of synovial fluid. Natural chitosan is a naturally available cationic glycan that functions similarly to HA and can act as a bio-lubricant for treating arthritis. The clinical lubrication properties of KiOmedine^®^ CM-chitosan, a non-animal carboxymethyl chitosan, have been evaluated by Vandeweerd et al. [29]. In vitro tribological experiments showed that this chitosan significantly reduced the COF due to the lubricating ability of the cross-linked HA formulations. In addition, chondroitin sulfate with glucosamine has also been used as a biological lubricant, which is commonly used clinically for arthritis relief and treatment [30,31]. Synthetic bio-lubricants have also shown good performance in terms of the lubrication and therapeutic effects. Through the ring-opening disproportionation polymerization of methyl 5-oxonorbornene-2-carboxylate, Wathier et al. [32] synthesized a polyanionic bio-lubricant (Figure 5). Friction experiments have shown that the polymers with low molecular weight showed a lower COF and significantly enhanced the viscosity of synovial fluid compared to saline, Synvisc, and bovine synovial fluid (BSF). Inspired by the bottle-brush structure possessed by biomolecules, Hartung et al. [33,34] prepared a series of brush lubricants with poly-L-lysine (PLL) or polyallylamine (PAAm) as the main chain and flexible PEG as the side chain, and their lubricating properties were also related to the length of the PEG chain and the grafting density. The PLL or PAAm can be bonded to negatively charged surfaces by electrostatic interactions to form a boundary lubrication layer [35]. Pettersson et al. [36] copolymerized PEO_45_MEMA with methacryloxyethyl trimethyl ammonium chloride to obtain a new type of bio-lubricant, which can also form a boundary lubrication layer on the substrate surface through electrostatic interaction, and the lubrication performance is mainly determined by its chain density.

While synovial fluid provides good lubrication as a fluid environment, the articular cartilage plays a more critical role in lubrication. The synovial joints of the human body are covered with a thin layer of articular cartilage (1–3 mm thick), which has a sponge-like macromolecular network structure. The synovial fluid’s water will penetrate the network during the sliding process, while charged water-soluble biomolecules can be assembled onto the cartilage surface to realize boundary lubrication [37]. The surface of the cartilage is also covered with HA, polyproteoglycans, and lubricin. The size of these macromolecules creates a site-barrier effect, and their strong hydration capacity allows them to freely extend into the solution and form a hydration layer [38]. This stable and dense layer has good adhesion and hydrated fluidity, allowing it to withstand high loads while maintaining a low friction factor. Inspired by the human joint lubrication mechanism, polymer brushes have been utilized to obtain superior tribological properties and good biocompatibility by grafting them onto desired surfaces to achieve functional mimicry of the joint lubrication system. Surface-grafted biomimetic polymer brushes mainly refer to the grafting of polymers from or onto surfaces by physical adsorption or covalent bonding, with the hydrophilic portion at the other end having no or only weak forces with the substrate. When the polymer chains are densely distributed, spatial repulsion causes the polymer to elongate and form a dense polymer brush layer of a certain thickness on the surface of the substrate [39]. In aqueous environments, the polymer brushes have a high penetration pressure and thus exhibit excellent lubricating properties with high load carrying.

Based on the inherent lubricating properties of natural polymers in living organisms, natural polymers were first modified, and their tribological properties were investigated. By functionalizing the natural macromolecules present in synovial joints, such as hyaluronic acid [40,41,42], polysaccharide [43,44,45], and phospholipid [46,47,48], researchers well modeled the tribological properties of mammalian joint lubrication systems. Based on the promoted understanding of the hydration lubrication mechanism of articular cartilage surfaces, the researchers further synthesized various types of cartilage-mimicking surface lubrication materials by various methods, such as surface-grafted polymer brush layers, surface-adsorbed polymer brush layers, and gel matrices with intrinsic surface lubrication [49].

In addition to the use of polymer brushes to achieve efficient lubrication, by adjusting the external conditions to apply stimuli to the lubrication layer, such as solvent [50], light [51], temperature [52], pH [53], electric field [54], and shear stress [55], the conformation of some polymers can be changed accordingly to achieve further modulation of the interface lubricating properties. For example, based on the mimicry of the lubrication performance of fish skin, Wu et al. [56] further introduced the pH-sensitive monomers sodium methacrylate (NaMA) and 2-(dimethylamino)ethyl methacrylate into the temperature-sensitive graphene-pNIPAM gel system, obtaining a hydrogel with the dual responsiveness of the pH and temperature (Figure 6a–d). The hydrogel has an ultra-low COF (≈0.05), which can be gradually varied from 0.05 to 1.2 by sequentially adjusting the pH and temperature of the solution reversibly, without structural damage to the gel. Wang et al. [57] prepared semi-transformable hydrogels with reversible photo-responsive supramolecular lubrication properties by integrating a responsive supramolecular system of α-cyclodextrin/poly(ethylene glycol) (α-CD/PEG) and a competing guest, 1-[p-(Phenylazo)benzyl]pyridinium bromide (AzoPB), into the frameworks of poly(vinyl alcohol) (PVA) and PAAm. Upon irradiation using UV and visible light, respectively, the competitive host–guest interactions between the α-CD/PEG supramolecular network and AzoPB led to the repeated formation and disappearance of sol–gel layers on the surface of the hydrogels, whereas the PVA and PAAm were unaffected and maintained their backbone properties, thus providing a reversible photo-responsive lubrication capability with variable toughness (Figure 6e,f). Inspired by the mechanism of transition from lubrication to astringency in the oral environment, Deng et al. [58] simulated this transition from ultra-low friction to a high friction state by combining mucin with PVA and achieved a large span of lubrication state switching (μ~0.009 to μ~0.47) by the interactions between mucin and tannic acid (Figure 6g–i).

Inspired by the lubrication mechanism in living organisms, the modulation of friction through the chemical properties of surfaces, as exemplified by polymer brushes, can fundamentally regulate the lubricating properties by controlling the degree of hydration to change the molecular state of the surface and achieve a significant reduction or reversible modulation of the COF in aqueous environments, which has brought great convenience and manipulability. However, most of the strategies for modulating interfacial interactions through interfacial chemistry find it difficult to take into account the surface roughness, hardness, deformability, and other factors that may result in a non-ideal contact state under real conditions, which may lead to a significant reduction in the lubricating performance of the material under real conditions. In addition, for friction modulation systems with stimulus-response capability, the surface’s molecular state or the response layer’s size limits the magnitude of the lubrication regulation. In addition, the actual application environment is far less stable than in the laboratory, and the required conditions imposed in the response process may be difficult to realize precisely in real use.

## 3. Surface Structure-Dominated Friction

The successful application of the surface structure in tribological performance optimization dates back to the 1940s, and surface geometry has also been extensively studied as an essential influence in tribology, in addition to interfacial chemistry [59,60]. Researchers have long noted that many organisms in nature have evolved various types and scales of surface structures to significantly change the tribological properties to adapt to complex living environments. We can find many examples in nature, such as lotus leaves [61], gecko toe pads [3], shark skin [62], and snake skin [63], where the structures of different surfaces confer different tribological properties (Figure 7a–f). Accordingly, researchers have designed a variety of surface-structured arrays to modulate the contact condition at the interface, thus obtaining tribological properties similar to those of biological surfaces.

For example, the lotus, one of the most famous organisms with superhydrophobic surfaces in nature, has attracted the attention of biologists and materials scientists since the last century and has been extensively studied in the field of drag reduction at solid–liquid interfaces [64]. The surface of the lotus leaf is rough and randomly distributed with many microcapillaries with branching nano-stratified structures at the top of the papillae. Thus, an air-lubricated membrane layer can be formed between the solid phase surface and the liquid phase due to the combined effect of the micropapillary structure and epidermal waxes [65]. The lubrication reduces the frictional resistance at the air–liquid interface, which allows the water droplets to roll easily on the surface of the leaf [66]. Bidkar et al. [67] further demonstrated the drag reduction capability of this type of hydrophobic surface by preparing randomly textured surfaces on flat plates and performing turbulence experiments. The skin-friction resistance was reduced by 20~30% in the experiments. Inspired by the surface structure of the lotus leaf, researchers have also prepared various surfaces with micro- and nano-graded structures, which have been widely used in waterproofing [68], ice-proofing [69], and self-cleaning [70,71].

Shark skin is also a rough surface capable of providing less frictional resistance, with oriented ribs of ordered size and space covering the shark’s dermis [72]. The rib-like teeth of the skin are present as grooves along the direction of the water flow, which reduces the friction between the water and the surface of the shark’s skin by decreasing the intensity of the turbulence [4]. At the same time, the interstices between these grooved structures also reduce the adhesion of the surface, making it difficult for tiny aquatic organisms to adhere to the shark’s body. Inspired by the shark skin structure, researchers have conducted a series of studies on this type of drag-reducing surface. Berchert et al. theoretically investigated the effect of several types of rib geometries on drag reduction, providing theoretical guidance for subsequent designs [73]. Shark skin-inspired rib structures have been demonstrated to reduce drag by up to 9.9% [74]. Xing et al. [75] prepared bionic shark skin textures with DLC coatings on Si_3_N_4_ ceramic. The sample exhibited a COF of 0.21 at 300 °C, which was 37.26% lower than that of the blank ceramic. Qin et al. [76] investigated the friction behavior of soft materials by preparing a bionic shark texture on polydimethylsiloxane (PDMS). Based on the synergistic effect of the bionic aligned texture and plasma treatment, the friction on the PDMS surface was effectively reduced. In addition, this type of structure has also been used in various applications, such as fluid drag reduction and antifouling [77].

As a limbless reptile with an elongated body covered with scales, snakes rely on friction between their body and the ground for locomotion [78]. This type of locomotion requires that the scales on their body surfaces generate sufficient friction to support the forward movement of the body but also provide a low coefficient of friction when the body is sliding. Researchers have studied the tribological properties of snakes’ body surfaces in different locomotion states and found that snakes exhibit significant anisotropy when moving in other directions. The COF was higher when the snake moved in the other direction and 1/4 to 1/2 of the other direction when moving forward [79,80]. The snake’s scales have a multiscale surface structure, with fibrous structures constituting micrometer-scale fiber waves with asymmetric tips. During changes in a snake’s state of locomotion, the interface between the fibers and the ground constantly changes between the tips and the lateral, causing the contact area to change, resulting in the snake’s body surface displaying different friction coefficients in different motion directions. The regulation mechanism of the snake’s skin originates from the variations brought about by the multilayers of the surface structure and asymmetries in the contact interfaces. The researchers have already achieved drag reduction and the lubrication effect by mimicking and optimizing this microstructure in the wet and dry state and on various organic and inorganic surfaces [81,82,83,84].

**Figure 7 biomimetics-09-00200-f007:**
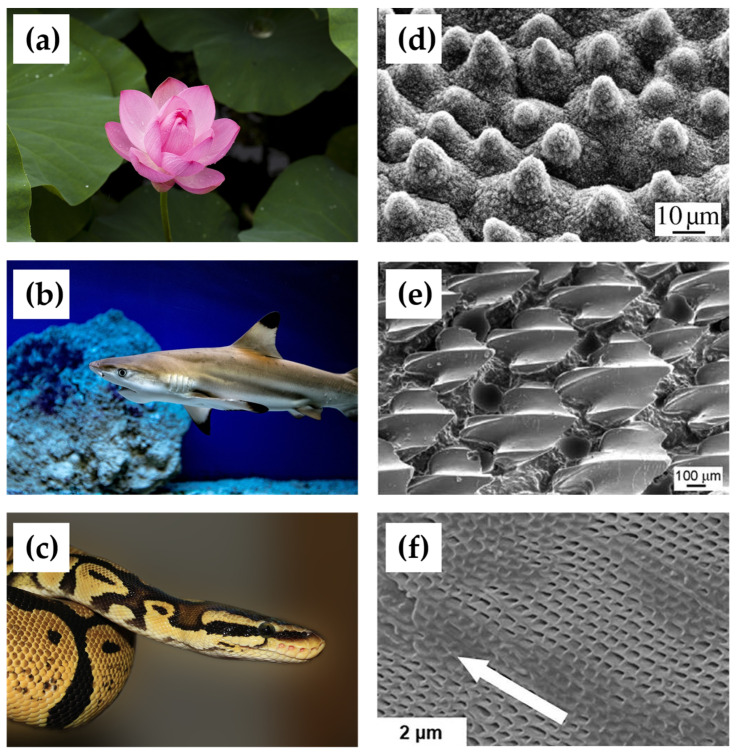
Images of a (**a**) lotus leaf, (**b**) shark, and (**c**) snake and the corresponding surface structures. (**d**) Papillae on the surface of the lotus leaf [85]. Copyright Permission from ACS, 2004. (**e**) Grooves on the scales of the shark [86]. Copyright Permission from Wiley, 2013. (**f**) Periodic structures on the body surface of the snake [83]. Copyright Permission from Springer Nature, 2023.

With the wide variety of plants and animals in nature, researchers have developed a variety of biomimetic lubrication materials with unique tribological properties inspired by various surface structures, either through alternative approaches or by focusing on their strengths. For example, based on the microstructure of the head of the dung beetle, You et al. [87] developed a structured surface that reduces friction by decreasing the contact area and trapping abrasive particles, and its resistance to cutting decreased by 30.41% compared with conventional materials. The gill covers of water snails continuously rub against their hard shells without significant wear. Xu et al. [88] revealed the fluid lubrication mechanism by observing and numerically analyzing the microgroove structure on the gill cover surface, which provides a COF as low as 0.012 in a liquid environment. Gregory et al. [89] studied the low resistance structures of rice blades and butterfly wings and coated nanostructures with the lotus effect onto polyurethane products with shark skin structures. The composite surfaces successfully mimicked the functions of rice blades and butterfly wings, advancing the understanding of surface design elements of biomimetic structures.

Currently, the means of obtaining the surface structures of materials include 3D printing technology and photolithography, which are greatly affected by the manufacturing cost, process precision, and time required. A large part of the material is still in the stage of laboratory preparation in small quantities, making it difficult to realize large-scale industrial preparation. For rigid substrates, the excessive load in the loading, friction, and unloading process is prone to cause severe damage to the structure, which leads to a decline in tribological performance and to lubrication failure; for soft substrates, the deformation after loading will also have an impact on the actual state of the surface structure during the friction process. Especially for systems that require the adaptive adjustment of the tribological performance, it is often difficult to achieve satisfying and continuous lubrication in complex working conditions by relying on only the surface structure to reduce friction.

## 4. Mechanics-Dominated Friction

Many organisms in nature have evolved surfaces with unique tribological properties and, at the same time, functional organs with specific or adjustable mechanical properties to maintain the adaptive working status under extreme conditions or to switch working states rapidly [90,91]. For example, sea cucumbers can escape danger by hardening the dermis to achieve sudden changes in surface stiffness, and many fish can escape from their captors by contraction hardening and deformation of the muscles [92,93] (Figure 8a,b). The mechanical properties of material surfaces greatly influence the contact state of the surface interface and directly affect the total friction force [94]. Researchers have long been concerned with deformation due to differences in the surface mechanical properties when studying elastomers such as rubber and the significant effect of hysteresis and loss on the total friction. In studies on the tribological behavior of human skin, friction brings about a large amount of lateral deformation, and the contribution of deformation friction to the total friction can be close to 50% at high speeds [95,96] (Figure 8c). For the mammalian joint system, the orderly hierarchical fibrous structure of nano/micro-collagen fibers endows the articular cartilage with excellent mechanical properties, which allows the shear forces in joint motion to be well carried and dispersed, thus cooperating with the synovial fluid and hydrophilic polymer layer on the cartilage to provide long-lasting adaptive lubrication [97,98] (Figure 8d).

In the friction lubrication system described above, the surface mechanics of the material greatly determine the contact state between the biological surface and the target substrate. If we can further modulate the mechanical properties based on considering the chemical properties and structure of the surface, further optimization of specific tribological properties or inducible switching of lubrication states can be achieved. Materials that change surface/subsurface stiffness in response to stimuli have been used in soft actuators and soft robotics research [99,100,101]. Modulating friction and lubrication performance via changes in the surface mechanical properties is easier for engineering applications than materials that modulate friction through interfacial chemistry. However, obtaining good friction and lubrication properties is difficult when relying on only a single change in mechanical properties without structuring or chemically treating the material’s surface.

## 5. Multiple Strategies Coupling-Dominated Friction

Many lubricating materials and devices based on a single biomimetic design strategy have been reported. However, obvious functional limitations still make it difficult to fully meet people’s production and life needs. Some of the materials remain at the stage of conceptual design and laboratory validation, and it is not easy to advance to the level of actual technological transformation. Therefore, it has become a hotspot and a challenge to study and understand the friction control mechanism of biological organs in nature from multiple perspectives and to develop high-performance or intelligent materials by coupling the design strategies of interfacial chemistry, surface structure, and surface mechanics.

### 5.1. Surface Chemistry Coupling Structure

The strategy of coupling the interfacial chemistry and surface structure enables friction reduction by reducing the contact area through the surface structure and further enhancing the lubrication effect through chemicals on the surface. For example, plants such as the Nepenthes pitcher plant use the structure on the surface to lock in the mucus it secretes [102]. Through an excellent match of solid and liquid surface energies, coupled with the roughness due to the microstructure, the surface can form a stable and effective liquid film that allows errant insects to slide down [103,104]. Wong et al. [105] were the first to introduce the concept of a slippery liquid-infused porous surface (SLIPS) and prepared surfaces with excellent stability, liquid repellency, and adhesion resistance using inexpensive materials such as Teflon. Ma et al. [106] used a simple nanosecond laser treatment method to prepare SLIPSs on carbon steel substrates. In addition to excellent hydrophobicity and corrosion resistance, the tribological properties of the smooth surfaces were improved, with the COF decreasing from about 0.52 to about 0.13 for the base steel. Tong et al. [107] further prepared a smart SLIPS coating inspired by the mucus-secreting behavior of the blind eel. Based on the responsive supramolecular interactions between azobenzene and α-cyclodextrin, the surface could achieve self-replenishment of the lubricant on the surface by contraction of the polymer chains under visible light or thermal stimulation.

The epidermal friction reduction of earthworms is also based on the synergistic effect of mucus secreted by their epidermal glands and annular grooves on the body surface [108,109]. The mucus secreted by the glands forms a lubricating layer on the earthworm’s body, while the grooves store the mucus and keep the lubricating layer stable while forming a gap between the body and the soil. Zhao et al. [110] mimicked the lubrication mechanism of earthworms and introduced textured structures onto the liquid-releasing polymer coatings, and the lubricants were stored as discrete droplets in a supramolecular matrix prepared from urea and polydimethylsiloxane copolymers. When the rough surface is subjected to localized pressure, the lubricant is released from the matrix and covers the corresponding area, achieving self-replenishing lubrication. Ruan et al. [111] combined the advantages of porous polyimide and phase change materials by impregnating paraffin wax into the porous material. They constructed smart lubrication materials with the ability to self-repair the lubrication layer. The material can release the internal lubricant under thermal stimulation and form a new paraffin lubrication layer on the surface quickly after the original layer is worn out. This type of coupling strategy can optimize the contact condition of the interface to some extent and improve the interfacial interaction, as well as optimize the stability and continuity of the lubrication layer. However, for solid lubrication, most interfacial chemical interactions are complicated to regulate and require specific means to immobilize the corresponding molecules onto the structured surface, making it difficult to achieve stable and rapid preparation in practical applications.

### 5.2. Surface Chemistry Coupling Mechanics

Combining the surface chemistry with the surface mechanics can lead to materials with outstanding performance through specific surface modification and substrate stiffness selection, as well as realize large-span lubricating state switching through the change in mechanical properties. The superior lubrication performance of mammalian articular cartilage is attributed to the dense and stable hydrophilic macromolecular layer on its surface and the well-organized layered structure with excellent adaptive load-bearing capacity. Inspired by the lubrication mechanism of articular cartilage, researchers have designed and synthesized a variety of high-performance propriety polymer lubrication materials [112,113,114] and surface-modified polymer lubrication materials [115,116,117,118], aiming to realize the effective combination of surface lubrication and propriety load-bearing of real cartilage. In addition, researchers have also achieved substantial tuning of the lubricating properties by hydrating the lubrication layer with a responsive substrate. Liu et al. [119] reported a temperature-responsive layered material prepared by brush-grafting the poly(potassium salt of 3-sulfopropyl methacrylate) onto the sub-surface of an initiator-embedded, high-strength hydrogel [poly(N-isopropylacrylamide-co-acrylic acid-co-initiator/Fe^3+^)] [P(NIPAAm-AA-iBr/Fe^3+^)]. The soft hydrogel/brush on the top layer provides hydration lubrication, and the temperature-sensitive hydrogel layer at the bottom provides adaptive load-bearing capacity, exhibiting tunable mechanical properties in response to temperatures above or below the lower critical solubilization temperature (LCST) (Figure 9a,b). Fish exhibit unique locomotion and lubrication mechanisms based on a highly hydrated body surface with modulus-adaptive muscle enhancement. Zhang et al. [93] proposed a modulation strategy for interfacial lubrication control based on modulus changes. The modulus-adaptive lubrication hydrogel (MALH) consists of a hydrophilic lubrication layer at the top and a thermally hardened phase-separated layer at the bottom, in which the bottom hydrogel can change from a soft state (20 °C, modulus of elasticity ~0.3 MPa) to a rigid state (80 °C, modulus of elasticity ~120 MPa), which enables the material to achieve switchable lubrication states in water when heated (COF from ~0.37 to ~0.027) (Figure 9c–g). The researchers further designed the Modulus Adaptive Switching Lubrication Device (MASLD) and demonstrated the promising application of this regulatory strategy in flexible devices and smart lubrication systems. The above strategy optimizes the lubrication performance through further knowledge and understanding of the lubrication mechanism of biological organs. The coupling of the two possible means of responsive modulation makes substantial tribological performance tuning possible. However, most of the materials studied so far are limited to single-component externally stimulated modulation, and it is not easy to realize synergistic modulation between the lubrication layer and the substrate material, which makes it difficult to realize a wide range of applications.

### 5.3. Simultaneous Coupling of Three Strategies

Considering the advantages and disadvantages of the above two design strategies, we can combine the three previous single strategies to develop novel biomimetic lubricating materials. Interfacial chemistry provides specific interaction force properties and regulatory mechanisms, surface structure provides optimized contact states, and surface mechanics provide the desired load-bearing capacity and dynamically tunable response states. Zhang et al. [120] proposed a method to synthesize a large-span viscous-slip switchable hydrogel by combining dynamic multiscale contact and coordinate regulation, which can achieve temperature-responsive viscous-slip switching. The responsive process mainly consists of molecular-scale chemical modulation that mimics the adhesion mechanism of mussels and mesoscale modulation based on surface roughness and modulus changes (Figure 10a–c). This smart hydrogel (DMCS-hydrogel) with dynamic multiscale contact synergistic modulation can be applied to various substrate surfaces and exhibits fast switching capability. Considering the coupled design of the three factors, Liu et al. [121] created a biomimetic high-strength anisotropic layered lubrication hydrogel (ALLH) with an ultra-low COF by coupling a hydrophilic polyelectrolyte brush, an anisotropic surface microstructure inspired by scallion leaf, and a high-mechanical-strength substrate mimicking human cartilage (Figure 10d–g). The artificial scallion leaves exhibit low friction (COF < 0.01) in different sliding directions under a wide range of contact stresses (≈0.2 to 2.4 MPa, corresponding to loads of ≈5 to 60 N) and ultra-low friction (COF ≈ 0.006) along the microstrip structure. For high contact pressure and long-term durability tests, the material achieves almost zero surface wear, which mimics human cartilage’s physiological function.

By coupling the three strategies, the problems of the single chemical property of the material itself and the unsatisfactory interfacial contact state are improved. However, the more factors that are combined, the more complicated the process links that need to be considered and the more parameters that to be regulated during the material preparation. Considering the laboratory operation limitations, most current materials and devices are multi-material composites, and the performance differences between different materials and the weak interfacial bonding remain to be solved. In the future, integrating the advantages of various materials and developing propriety functional materials with excellent performance or easy modification to realize the on-demand design and manufacturing of bionic lubrication materials will remain a significant challenge.

## 6. Summary and Perspective

With people’s deepening understanding of the mechanism of biological lubrication, a variety of biomimetic lubricating materials with better design strategies have been reported one after another. The single biomimetic lubrication strategy has been widely used in developing practical and functional lubricating materials. Polymer brush systems inspired by articular cartilage have made a big splash in water lubrication systems and bio-lubrication, while structuring processes based on animal and plant surface structures have been widely used in self-cleaning, fluid drag reduction and antifouling. The comprehensive influence of surface mechanics on the friction or lubrication performance of materials has also been gradually emphasized by researchers.

However, most of the development of bionic lubrication materials and devices is still limited to the laboratory stage. It is difficult to meet the harsh conditions of use, which puts forward new practical requirements. Given the inadequacy of a single biomimetic strategy, researchers have begun to develop biomimetic materials by coupling multiple factors. Surface modifications such as hydrophilicity/hydrophobicity can further optimize the properties of structured surfaces produced by conventional processes. In contrast, the surface structure, in turn, improves the contact state of the host material or optimizes the durability and stability of tribological properties. In addition, by introducing the factor of the surface/subsurface mechanical properties, the lubrication state is expected to be further optimized and drastically regulated.

Coupling strategies can compensate for the shortcomings of a single strategy to a certain extent while highlighting its advantages and maximizing the utility of each mechanism. However, multiple regulatory factors often bring about more complex design strategies and manufacturing processes, and researchers often need to integrate and regulate the performance of multiple functionalized systems. The differences in properties of various materials can easily bring about insufficient bonding power and difficulties in regulation. In this case, how to reasonably couple the advantages to obtain a responsive propriety functionalized material that is easy to regulate or develop a composite material with better performance is still a great challenge. From the engineering point of view, to ensure the efficient, continuous and reliable lubrication performance of materials or devices under complex and harsh conditions, realizing adaptive lubrication performance under real and variable working conditions is also a major focus and difficulty. With the deepening research and understanding of the interfacial lubrication mechanism in the biological movement process and the continuous innovation of the material synthesis process, the development of new biomimetic friction lubrication materials with the ability to adapt to the working conditions or environment will become one of the critical development directions in the field in the future, with a focus on the adjustable interfacial contact state, through the combination of polymer design and synthesis, multiscale surface structuring, surface mechanical property regulation and mechanical deformation, and so on. In the future, these materials are expected to shine in biomedicine, intelligent electronic sensor devices, soft robots, and precision manufacturing.

## Figures and Tables

**Figure 1 biomimetics-09-00200-f001:**
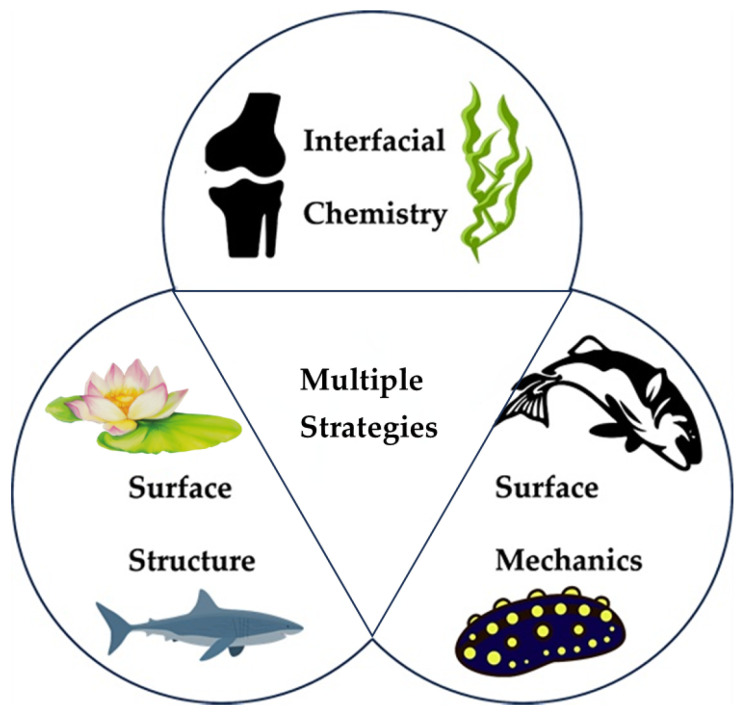
The schematic shows the three biomimetic strategies for achieving friction control.

**Figure 2 biomimetics-09-00200-f002:**
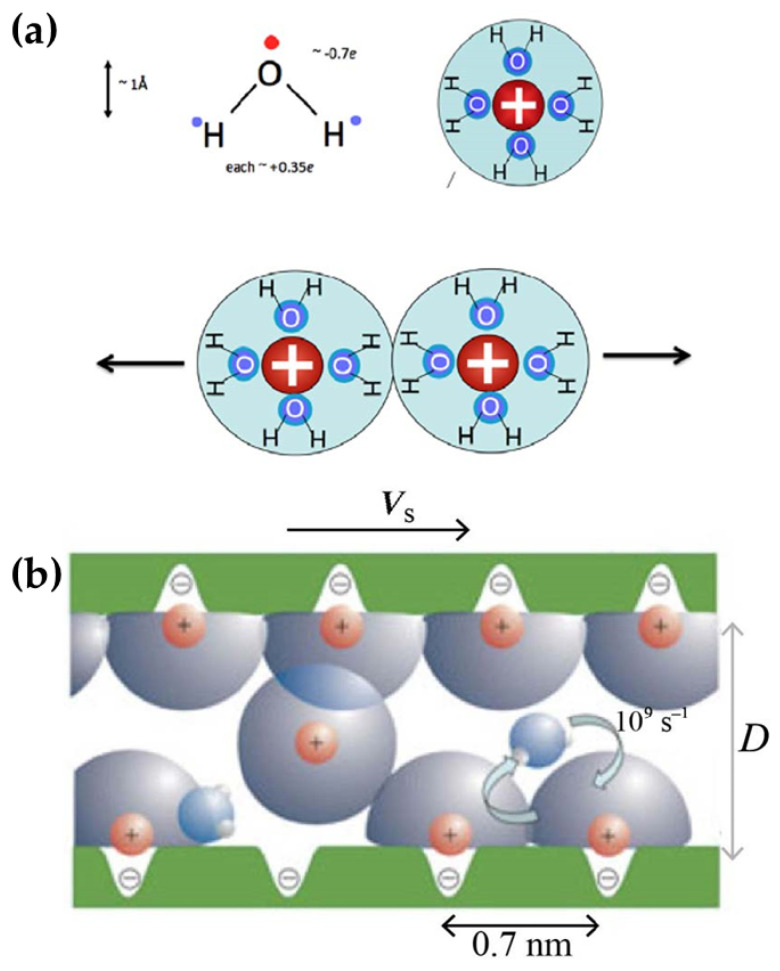
(**a**) The large dipole of water and the formation of hydration shells about charges. (**b**) The mechanism of hydration lubrication between charged surfaces across trapped hydrated ions [7]. Copyright Permission from Springer Nature, 2013.

**Figure 3 biomimetics-09-00200-f003:**
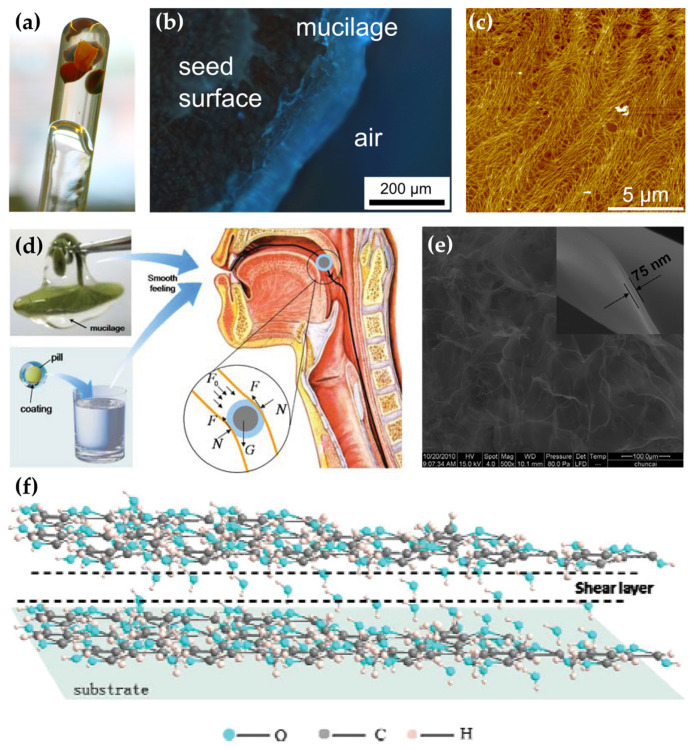
(**a**) Photograph of gel-like mucus obtained from papaya seeds, (**b**) gel-like layer formed on the surface of the seeds after being dissolved for 20 min using calcium fluoride solution, and (**c**) AFM morphology image of fresh papaya mucus aggregated on a mica sheet [13]. Copyright Permission from Elsevier, 2014. (**d**) Brasenia schreberi mucilage and its lubrication, (**e**) SEM image of Brasenia schreberi mucilage after treatment by the vacuum freeze-drying method, and (**f**) schematic of polysaccharide nanosheets in mucilage during lubrication [15]. Copyright Permission from ACS, 2012.

**Figure 4 biomimetics-09-00200-f004:**
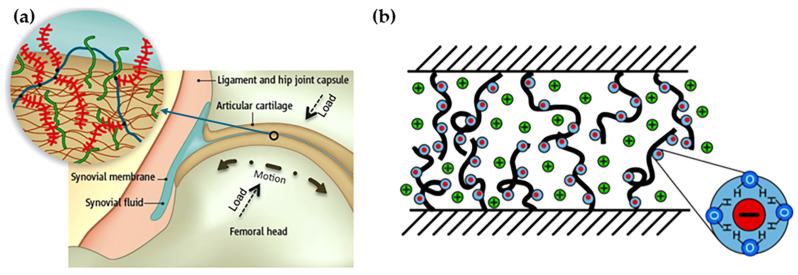
(**a**) Illustration of the natural articular cartilage system and the functional biomolecules in it: HA (blue), polyproteoglycan (red bottle-brush molecule), and lubricin (green) [21]. Copyright Permission from AAAS, 2009. (**b**) Highly hydrated phosphorylcholine groups are a highly effective lubricating element, and the figure illustrates the hydrated phosphorylcholine headgroups exposed on the surface of the liposomes as they slide relative to one another [27]. Copyright Permission from ACS, 2015.

**Figure 5 biomimetics-09-00200-f005:**
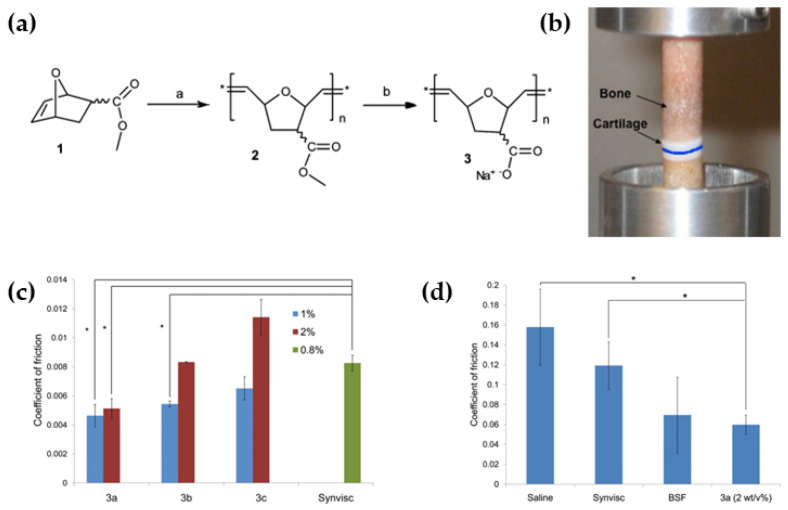
(**a**) Structural formula for Poly(7-oxanorbornene-2carboxylate) and (**b**) schematic diagram of its lubrication test model. (**c**) Polymer 3a with the lowest molecular weight has the lowest COF, (**d**) and exhibits lubricating properties superior to those of BSF [32]. Copyright Permission from ACS, 2013.

**Figure 6 biomimetics-09-00200-f006:**
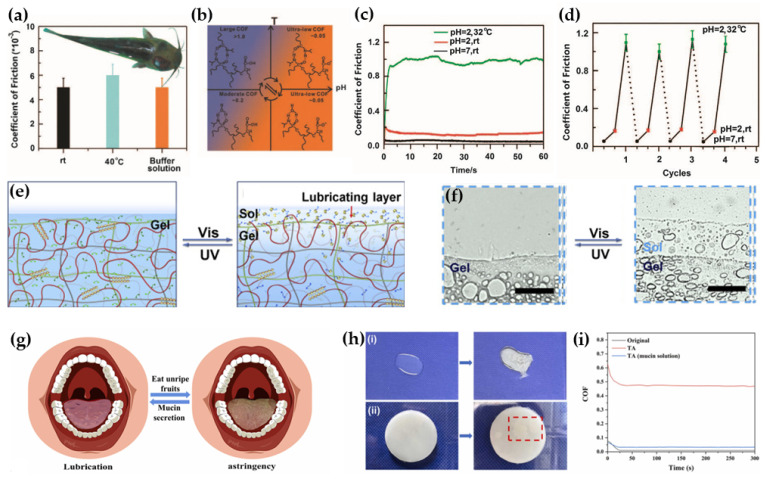
(**a**) Images of catfish skin under different conditions and the corresponding COF, (**b**) pH and temperature changes induce a reversible swelling–collapse cycle, (**c**) the COF curves of the pNIPAM_11_–NaMA_3_ gel at pH = 7 and 2, rt, and pH = 2 at 32 °C, and (**d**) the switchable COF of the pNIPAM_11_–NaMA_3_ gel with the stimuli of the pH and temperature [56]. Copyright Permission from Springer Nature, 2014. (**e**) Schematic of the possible photo-responsive lubrication mechanism of PSCHs, (**f**) images of the sol–gel transition in UV and visible light [57]. Copyright Permission from Elsevier, 2021. (**g**) Schematic representation of the switchable lubrication behavior in the oral cavity, (**h**) interaction of mucin solution and PVA/mucin hydrogel with tannins, (**i**) COFs of pristine hydrogel (black line), TA-treated hydrogel (red line), and TA-treated and incubated hydrogel in mucin solution (blue line) [58]. Copyright Permission from Elsevier, 2023.

**Figure 8 biomimetics-09-00200-f008:**
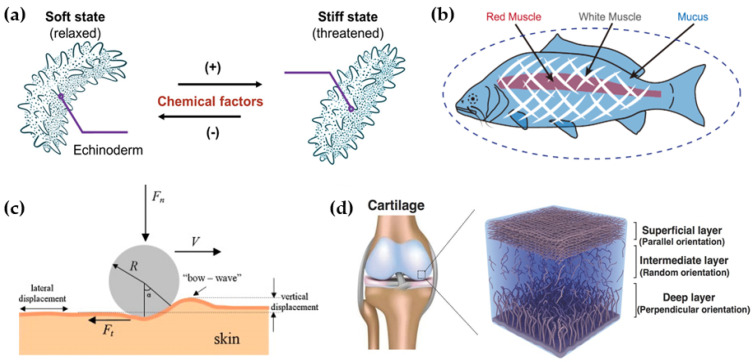
Schematic representation of (**a**) a sea cucumber [92] and (**b**) a fish [93]. The sea cucumber reversibly transforms its dermal hardness when threatened, while the fish escapes by struggling with epidermal mucus. Copyright Permission from Wiley, 2022; Copyright Permission from Springer Nature, 2022. (**c**) Schematic representation of the deformation of the skin surface during friction [95]. Copyright Permission from Elsevier, 2009. (**d**) Layered and organized structure of articular cartilage [98]. Copyright Permission from Wiley, 2017.

**Figure 9 biomimetics-09-00200-f009:**
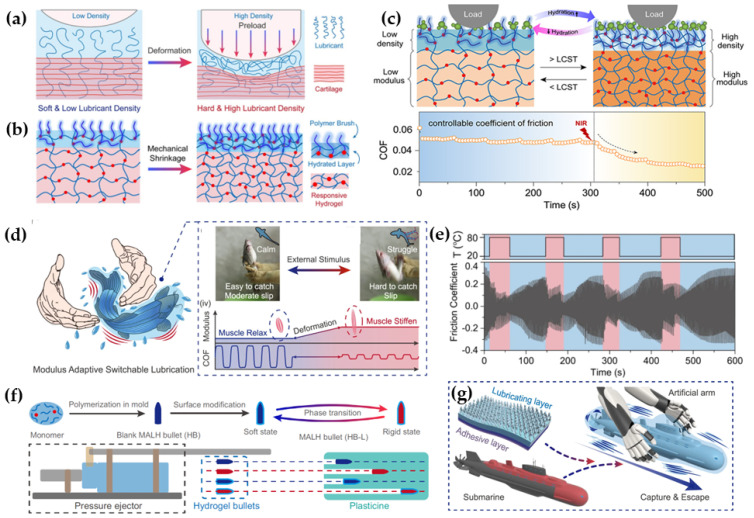
(**a**) Schematic representation of adaptive mechanically controlled lubrication in the joint during unloading and loading; (**b**) schematic representation of the change in density of the top hydrogel/brush composite layer and the change in the mechanical strength of the underlying hydrogel layer as the material network shrinks during heating; (**c**) improvement of the interfacial lubrication of this hydrogel during the heating process, which exhibits dynamic adaptation [119]. Copyright Permission from ACS, 2020. (**d**) Struggling behavior of a fish during capture and its skin muscle modulus versus COF; (**e**) evolution of the COF during in situ heating and cooling of the MALH; (**f**) demonstration of the MALH as a smart bullet; (**g**) schematic diagram of the underwater in situ capture device of the MASLD [93]. Copyright Permission from Springer Nature, 2022.

**Figure 10 biomimetics-09-00200-f010:**
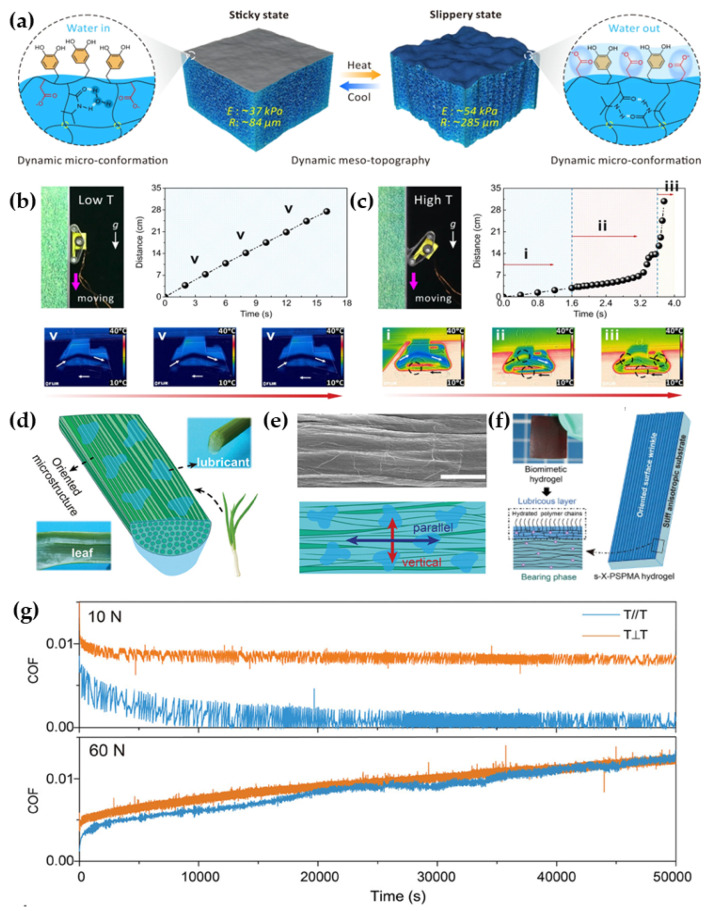
(**a**) DMCS-hydrogel based on dynamic multiscale contact synergy originating from the dynamic meso-topography and micro-conformation to realize the stick-slip switching; mobile device at low (**b**) and high (**c**) temperatures in the images and time–distance curves of crawling on a vertical metal plate, and infrared images showing the heat transfer process from the substrate to the DMSC-hydrogel [120]. Copyright Permission from Springer Nature, 2022. (**d**) Oriented distribution of substrate fibers and well-hydrated mucilage on the surface; (**e**) SEM images and schematic diagrams of the surface structure of scallion leaves; (**f**) schematic diagrams of ALLH samples; (**g**) the COF curves of the ALLH sample in the entire 50,000 sliding cycles under the normal loads of 10 N (contact pressure = 0.4 MPa) and 60 N (contact pressure = 2.4 MPa) in two perpendicular directions with water as the lubricant [121]. Copyright Permission from Wiley, 2024.

**Table 1 biomimetics-09-00200-t001:** Friction-reducing properties of the secretions of natural plants.

Creature/Tissue	Friction Substitutes and Velocity	COF	Reference
Aloe mucilage	WC ball/DLC flat; 150 mm·s^−1^	0.04	[12]
Papaya seed mucilage	Polyethylene flat/stainless steel flat; 100 mm·s^−1^	0.03	[13]
Red microalga secretion	Si_3_N_4_ ball/alumina flat; 0.2 mm·s^−1^	0.003	[14]
Brasenia mucilage	Glass flat/glass flat; 0.01 mm·s^−1^	0.005	[15]

## Data Availability

There were no new data created.

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
