# Peer review of "Bioinspired Interfacial Friction Control: From Chemistry to Structures to Mechanics"

_biomimetics, 2024, doi:10.3390/biomimetics9040200_

Round 1

Reviewer 1 Report

Comments and Suggestions for Authors

Does the authors agree that superhydrophobicity, e.g., learning from the self-cleaning of lotus where water has a very low friction on the surface, is a kind of bioinspired interfacial friction control?

I believe it is good to include some theoretical calculations, e.g., DFT/MD simulations/math theory in this review, because they also contributes to the field.

The anisotropy of chemical interactions can be also used to achieve friction control. Recent reports on anisotropy of π–π stacking as basis for superlubricity, and angle-dependent strength of a single chemical bond by stereographic force spectroscopy, revealed the interfacial interactions based friction control is also a good direction and can be included here.

Figure 3f seems blur. You can replace it with a higher resolution (if you can find).

The word "Figure" should be written in a unified way, for example, Fig 9c-9g, Figure 9a-9b.

Some of the references have to be unified. Either write the journal name all in abbreviation or all in full name. For example, Ref.51 is Chem. Coummun., while Ref.53 is Advanced Materials.

Some of the references lack full information, such as Ref.96 has no page number.

Comments on the Quality of English Language

English is good. Only minor editing of English language required.

Author Response

Dear Editor and reviewers:

Please find the revised manuscript entitled “Bioinspired Interfacial Friction Control: From Chemistry, Structures to Mechanics” (Manuscript ID: biomimetics-2908203) for possible publication in Biomimetics.

First of all, we would like to thank you for having our manuscript reviewed and give us a revision chance. Really many thanks to you. Also, all reviewers give us profound and inspiring comments. Their questions help us to further improve the quality of this manuscript. As a response, we have carefully revised the manuscript according to the comments from reviewers points by points, while the revised parts have been highlighted in red character. We hope that the reversion can convince you and the reviewers to accept our manuscript for publication in Biomimetics.

Thank you very much for your kind consideration of our manuscript.

With my best regards,

Sincerely,

Professor Shuanhong Ma

State Key Laboratory of Solid Lubrication

Lanzhou Institute of Chemical Physics

Chinese Academy of Science

Lanzhou 730000, China

Comment:

Q1: Does the authors agree that superhydrophobicity, e.g., learning from the self-cleaning of lotus where water has a very low friction on the surface, is a kind of bioinspired interfacial friction control?

Response: We deeply appreciate you for bringing up this question. In this paper, we focus on bio-inspired lubrication strategies and consider designs that mimic biological structures to achieve friction reduction as a form of interfacial friction control. In Section 2, we choose several biological surfaces in nature that possess lubricating properties as examples. Among them, lotus has good solid-liquid interface lubrication properties and the structure has also been used in solid lubrication in many areas. The multi-scale microstructure of the lotus leaf surface increases its surface roughness and reduces the real contact area, which is one of the typical examples of friction reduction through surface structure. At the same time, superhydrophobicity is one of the outward signs of its structural characteristics, rather than superhydrophobicity determining the tribological properties of the surface.

Q2: I believe it is good to include some theoretical calculations, e.g., DFT/MD simulations/math theory in this review, because they also contributes to the field.

Response: We deeply appreciate the reviewers for the suggestion. We fully believe the importance of methods such as numerical simulations and theoretical calculations in materials science and tribology research. In this paper, however, we focus on the inspiration provided by the diversity of organisms and their structures, and the application of interfacial friction control in solid lubrication through bionics. We are afraid that we can't add the appropriate theoretical calculations to the paper for now. We would like to add more theoretical calculations to the follow-up work.

Q3: The anisotropy of chemical interactions can be also used to achieve friction control. Recent reports on anisotropy of π–π stacking as basis for superlubricity, and angle-dependent strength of a single chemical bond by stereographic force spectroscopy, revealed the interfacial interactions based friction control is also a good direction and can be included here.

Response: We deeply appreciate the reviewers’ suggestion. Within the context of surface chemistry-dominated friction control, this paper focuses primarily on the lubrication mechanisms of living organisms. In contrast to diverse biomimetic strategies of properties such as adhesion, the most common lubrication effects in nature come from friction reduction due to hydration. We have carefully searched and learnt from the references suggested by the reviewers and have taken a closer look at the study of chemotactic anisotropy as a basis for superlubrication. In the study of pi-pi stacking model systems using single molecule force spectroscopy, the researchers demonstrated a deep understanding of adhesion and friction at interfaces based on molecular interactions. However, it is difficult for us to obtain similar ideas or strategies in biological structures. So we probably won't go further with the content in this paper. We are very interested in the application of single molecule force spectroscopy experiments in tribological studies based on AFM, and we would like to be able to refer to this type of paper in subsequent studies. We have gained a lot of useful information from your suggestion and thank you for the comment again.

Q4: Figure 3f seems blur. You can replace it with a higher resolution (if you can find).

Response: We are grateful for pointing out the problem in the pictures. However, we couldn’t get a clearer picture from the original article. We apologize for not being able to make a replacement for Figure 3f.

Q5: The word "Figure" should be written in a unified way, for example, Fig 9c-9g, Figure 9a-9b.

Response: We are grateful to the reviewers for pointing out the problem. The word “Figure” has been unified in the revised version of the manuscript.

Q6: Some of the references have to be unified. Either write the journal name all in abbreviation or all in full name. For example, Ref.51 is Chem. Coummun., while Ref.53 is Advanced Materials.

Q7: Some of the references lack full information, such as Ref.96 has no page number.

Response: Thanks for pointing out our mistakes. We have rechecked the formatting and completeness of all references and made corrections. The changes have been marked in the revised version of the manuscript.

Reviewer 2 Report

Comments and Suggestions for Authors

This paper reviews the bioinspired friction control studied. It summaries the cases in nature and the biomimetic studies related to the corresponding natural cases. The mechanisms to control friction is classified into three categories. I suggest the paper to be accepted after minor revision.

1. The review feels it is not appropriate to call the third mechanism as "Mechanics-dominated friction". Mechanics is a discipline that studies materials and structures and how they deform under load. The mechanism is about mechanical property, not mechanics. Since here mechanical property actually refers to stiffness, I will suggest calling it "stiffness modulated friction" or "stiffness controlled friction".

Author Response

Dear Editor and reviewers:

Please find the revised manuscript entitled “Bioinspired Interfacial Friction Control: From Chemistry, Structures to Mechanics” (Manuscript ID: biomimetics-2908203) for possible publication in Biomimetics.

First of all, we would like to thank you for having our manuscript reviewed and give us a revision chance. Really many thanks to you. Also, all reviewers give us profound and inspiring comments. Their questions help us to further improve the quality of this manuscript. As a response, we have carefully revised the manuscript according to the comments from reviewers points by points, while the revised parts have been highlighted in red character. We hope that the reversion can convince you and the reviewers to accept our manuscript for publication in Biomimetics.

Thank you very much for your kind consideration of our manuscript.

With my best regards,

Sincerely,

Professor Shuanhong Ma

State Key Laboratory of Solid Lubrication

Lanzhou Institute of Chemical Physics

Chinese Academy of Science

Lanzhou 730000, China

Comment

This paper reviews the bioinspired friction control studied. It summaries the cases in nature and the biomimetic studies related to the corresponding natural cases. The mechanisms to control friction is classified into three categories. I suggest the paper to be accepted after minor revision.

Q1:The review feels it is not appropriate to call the third mechanism as "Mechanics-dominated friction". Mechanics is a discipline that studies materials and structures and how they deform under load. The mechanism is about mechanical property, not mechanics. Since here mechanical property actually refers to stiffness, I will suggest calling it "stiffness modulated friction" or "stiffness controlled friction".

Response: We are grateful to the reviewers for the effort reviewing our paper and the positive feedback. We have carefully considered the reviewers' suggestions and re-examined the word choice for the paper's third subject heading. In contact mechanics, especially in the study of soft materials, when we talk about "mechanics", we are not only concerned with the surface/subsurface stiffness (modulus of elasticity) of the material, but also with the mechanical properties such as strength and toughness. In the study of materials with special surface/subsurface structures, such as gradient materials or porous materials, it is less than perfect to attribute only the change in mechanical properties when frictional properties are modulated to the control of stiffness. For most frictional systems in nature, research is based on soft-contact mechanics, so we are afraid it's difficult for us to change "mechanics" to "stiffness" in the paper. The reviewers' comments were very constructive and inspired us to think a lot, and we would like to thank you once again for your suggestion.

Reviewer 3 Report

Comments and Suggestions for Authors

The review article “Bioinspired Interfacial Friction Control: From Chemistry, Structures to Mechanics” shows the interesting research results in common bio-lubrication modulation in nature and the corresponding application of biomimetic materials in friction systems. Detailed studies of the mechanisms of biomodulation of interfacial friction from three perspectives, namely, interfacial chemistry, surface structure, and surface mechanics, respectively have been performed. However, some changes are needed to improve the manuscript.

1)      It is not immediately clear from Figure 4a in which area the enlarged fragment is located; it would be better to show it with an arrow.

2)      Section 2 presents a large body of data on biological structures that have excellent lubricating properties and reduce the coefficient of friction. These data should be combined into a table with the references.

3)      From the caption to Figure 5 it is not clear what is shown in (b) and (c); the caption for (d) is completely missing. There are no comments in the text for Figure 5.

4)      There is no scale bar in Figure 7d

5)      Contrary to section 2, section 3 presents too general information and does not provide specific examples from the literature of the use of biomimetic surface structures to reduce the coefficient of friction of materials.

Author Response

Dear Editor and reviewers:

Please find the revised manuscript entitled “Bioinspired Interfacial Friction Control: From Chemistry, Structures to Mechanics” (Manuscript ID: biomimetics-2908203) for possible publication in Biomimetics.

First of all, we would like to thank you for having our manuscript reviewed and give us a revision chance. Really many thanks to you. Also, all reviewers give us profound and inspiring comments. Their questions help us to further improve the quality of this manuscript. As a response, we have carefully revised the manuscript according to the comments from reviewers points by points, while the revised parts have been highlighted in red. We hope that the reversion can convince you and the reviewers to accept our manuscript for publication in Biomimetics.

Thank you very much for your kind consideration of our manuscript.

With my best regards,

Sincerely,

Professor Shuanhong Ma

State Key Laboratory of Solid Lubrication

Lanzhou Institute of Chemical Physics

Chinese Academy of Science

Lanzhou 730000, China

Comments:

The review article “Bioinspired Interfacial Friction Control: From Chemistry, Structures to Mechanics” shows the interesting research results in common bio-lubrication modulation in nature and the corresponding application of biomimetic materials in friction systems. Detailed studies of the mechanisms of biomodulation of interfacial friction from three perspectives, namely, interfacial chemistry, surface structure, and surface mechanics, respectively have been performed. However, some changes are needed to improve the manuscript.

Q1: It is not immediately clear from Figure 4a in which area the enlarged fragment is located; it would be better to show it with an arrow.

Response: We are grateful for the suggestion. We have added an arrow on Figure 4a in the revised version of the manuscript to make it clearer.

Q2: Section 2 presents a large body of data on biological structures that have excellent lubricating properties and reduce the coefficient of friction. These data should be combined into a table with the references.

Response: Thank you for the suggestion. As suggested by the reviewers, we have combined the data of hydration lubrication of plants into a table and adjustments have been made to the text of section 2.

Q3: From the caption to Figure 5 it is not clear what is shown in (b) and (c); the caption for (d) is completely missing. There are no comments in the text for Figure 5.

Response: We are grateful to the reviewers for pointing out the problem. We want to deeply apologize for this mistake. We have modified the caption of Figure 5 its counterpart in the main text, so that it could clearly conveys the meaning of the figure.

Q4: There is no scale bar in Figure 7d

Response: Thank you for pointing out the problem. A scale bar has been added to Figure 7 in the revised manuscript.

Q5: Contrary to section 2, section 3 presents too general information and does not provide specific examples from the literature of the use of biomimetic surface structures to reduce the coefficient of friction of materials.

Response: We deeply appreciate the reviewers’ suggestion. In Part 2, we describe in detail the hydration lubrication mechanisms that exist in nature and the corresponding bio-inspired friction systems and lubrication materials. The mechanisms of friction reduction and lubrication covered in this part are relatively homogeneous, so we hope to provide readers with an overview of this hot area of research through a detailed introduction. In Part 3, we select three representative biological surface structures and explain their friction reduction mechanisms, and briefly introduce their possible application areas. The paper focused on the diversity of species in nature and did not include many examples. In response to the reviewer's suggestion, we have added some examples of recent research in section 3.

Reviewer 4 Report

Comments and Suggestions for Authors

COMMENTS TO THE AUTHORS

The submitted paper focuses on interfacial friction control inspired by nature. The submitted review paper is well-written and contains only a few recommendations, which can be implemented in the revised version. Therefore, the submitted manuscript is recommended for publishing in the Biomimetics Journal after a minor revision.

1. The quality of Figures 3 E and F should be improved.

2. No need to use abbreviations which are not used again.

3. Slippery liquid-infused porous surface research could be included as well.

4. More examples of the latest trends in the research of biomimicking living creatures and species could be provided.

Author Response

Dear Editor and reviewers:

Please find the revised manuscript entitled “Bioinspired Interfacial Friction Control: From Chemistry, Structures to Mechanics” (Manuscript ID: biomimetics-2908203) for possible publication in Biomimetics.

First of all, we would like to thank you for having our manuscript reviewed and give us a revision chance. Really many thanks to you. Also, all reviewers give us profound and inspiring comments. Their questions help us to further improve the quality of this manuscript. As a response, we have carefully revised the manuscript according to the comments from reviewers points by points, while the revised parts have been highlighted in red. We hope that the reversion can convince you and the reviewers to accept our manuscript for publication in Biomimetics.

Thank you very much for your kind consideration of our manuscript.

With my best regards,

Sincerely,

Professor Shuanhong Ma

State Key Laboratory of Solid Lubrication

Lanzhou Institute of Chemical Physics

Chinese Academy of Science

Lanzhou 730000, China

COMMENTS TO THE AUTHORS

The submitted paper focuses on interfacial friction control inspired by nature. The submitted review paper is well-written and contains only a few recommendations, which can be implemented in the revised version. Therefore, the submitted manuscript is recommended for publishing in the Biomimetics Journal after a minor revision.

Q1: The quality of Figures 3e and 3f should be improved.

Response: We are grateful for pointing out the problem of the pictures. However, we can’t get a clearer picture from the original article. We apologize for not being able to make a replacement for Figure 3e and 3f.

Q2: No need to use abbreviations which are not used again.

Response: Thanks very much for your carefully revision. As a response, we have rechecked the abbreviations in the text and removed any unnecessary use.

Q3: Slippery liquid-infused porous surface research could be included as well.

Response: Thank you for your constructive comments. Slippery liquid-infused porous surface inspired by plants such as hogweed are a representative role in bionics. As a response, we have made corresponding additions in section 5.1.

Q4: More examples of the latest trends in the research of biomimicking living creatures and species could be provided.

Response: Thanks very much for your comment. We have added some recent papers in the field of bioinspired lubrication to the revised manuscript. The additions have been marked in the text and references.
